# Effects of sprint interval training compared to high intensity interval training on repeated sprint capacity and sport-specific performance in college-aged male tennis players

Jing Fan[1], Ke Sun [1]*, Xu Liu[2], Tianyu Zhu[1], Yue Li[3]

**1** Department of Physical Education, University of Chinese Academy of Sciences, Beijing, China,
**2** School of Physical Education and Health Engineering, Taiyuan University of Technology, Taiyuan, China,
**3** Physical Education Department, Shandong Jiaotong University, Jinan, China

* sunke@ucas.ac.cn

## Abstract

### Purpose

This study compared the effects of sprint interval training (SIT) versus high-intensity interval training (HIIT) on repeated sprint capacity and sport-specific performance in male tennis players.

### Methods

Thirty collegiate tennis players (age: $22.39 \pm 1.64$ years) were randomly assigned to SIT (n = 15) or HIIT (n = 15) groups for an 8-week training intervention. Performance and physiological parameters were assessed using repeated sprint ability (RSA) test, hit-and-turn tennis test, and tennis-specific endurance test. Key metrics included maximal completed level, total time to exhaustion (TTE), repeated sprint parameters (RSATT, RSAbest, Sdec), maximal oxygen uptake ($VO_2$max), ventilatory efficiency, and tennis performance index.

### Results

Significant group × time interactions were observed for maximal completed level ($F(1, 28)=22.649$, $p < 0.001$, $\eta p^2 = 0.89$), RSATT ($F(1, 28) = 39.74$, $p < 0.001$, $\eta p^2 = 0.89$), and ball accuracy ($F(1, 28)=13.811$, $p < 0.001$, $\eta p^2 = 0.901$). The SIT group showed greater improvements in maximal completed level ($4.00 \pm 0.28$ vs. $2.14 \pm 0.28$, $p < .001$), RSATT ($-4.84 \pm 0.33$s vs. $-1.87 \pm 0.33$s, $p < 0.001$), and ball accuracy ($12.85 \pm 1.71\%$ vs. $3.86 \pm 1.71\%$, $p < 0.001$). Both groups improved significantly in $VO_2$max (SIT: $52.84 \pm 4.22$ to $64.50 \pm 3.85$ mL.min$^{-1}$.kg$^{-1}$; HIIT: $51.79 \pm 5.15$ to $59.6 \pm 4.44$ mL.min$^{-1}$.kg$^{-1}$, $p < 0.001$) and TTE (SIT: $\eta p^2 = 0.825$; HIIT: $\eta p^2 = 0.59$). Time to the onset of blood lactate accumulation (OBLA) and second ventilatory threshold (VT2) showed significant main effects of time ($p < .001$) without group differences.

**Data availability statement:** All relevant data are within the paper and its Supporting information files.

**Funding:** The author(s) received no specific funding for this work.

**Competing interests:** The authors have declared that no competing interests exist.

## Conclusion

While both protocols improved aerobic fitness, SIT demonstrated superior effectiveness in enhancing tennis-specific performance, particularly in repeated sprint ability and technical stability. These findings suggest that SIT might be a more time-efficient training strategy for improving sport-specific performance in tennis players.

## Introduction

Physical conditioning in tennis has become increasingly important as the sport has evolved into a more demanding and dynamic game. Tennis players require a combination of aerobic and anaerobic fitness to perform effectively during matches that can last several hours [1]. The intermittent nature of tennis, characterized by short bursts of high-intensity efforts interspersed with brief recovery periods, necessitates specific training approaches to enhance both aerobic and anaerobic energy systems. High-intensity interval training (HIIT) and sprint interval training (SIT) have emerged as time-efficient alternatives to traditional endurance training for improving cardio-respiratory fitness and sport-specific performance [2]. While both training methods involve intermittent high-intensity efforts, they differ in their work-to-rest ratios and intensity levels. HIIT typically involves longer intervals (1–4 minutes) at submaximal intensities (85–95% maximal heart rate), while SIT consists of shorter, all-out efforts (15–30 seconds) with longer recovery periods [3].

Recent research has demonstrated that tennis-specific HIIT can improve various aspects of physical performance in tennis players. Fernandez-Fernandez, Sanz [4] found significant improvements in VO2peak and tennis-specific endurance following an 8-week HIIT intervention in young tennis players. Similarly, Kilit and Arslan [5] reported that both HIIT and on-court tennis training improved aerobic capacity and sprint performance in young tennis players. The effectiveness of SIT has been well-documented in various sports contexts. Turner, Pyne [6] demonstrated that both HIIT and SIT protocols led to significant improvements in performance parameters in national-level rowers. However, the specific application of SIT in tennis and its comparison to HIIT remains relatively unexplored. Durmuş, Ödemiş [7] conducted a systematic review highlighting the positive effects of HIIT on tennis players' aerobic capacity and technical abilities, but noted inconsistent results regarding agility, sprint, and jump performances.

The physiological demands of tennis —marked by frequent, high-intensity efforts— require training strategies that effectively target both aerobic and anaerobic energy systems. Suárez Rodríguez and Del Valle Soto [8] found that specific interval training protocols can enhance performance-related variables such as shot intensity and precision while mitigating fatigue in tennis players. Building on this, emerging research suggests that both HIIT and SIT may offer distinct but complementary benefits for tennis-specific conditioning. For instance, Hebisz, Cortis [9] found that a polarized training model, which combines HIIT and SIT elements, significantly improved cognitive-motor performance (e.g., choice reaction time) and physical performance

markers. These findings underscore the importance of exploring how different interval training modalities contribute to performance outcomes in tennis. Additionally, Kavaliauskas, Jakeman [10] found that early adaptations to SIT can differ from other high-intensity training protocols, suggesting potential sport-specific applications. The metabolic responses to HIIT and SIT have been shown to differ, which may influence their effectiveness for specific sport applications. Eigendorf, Maassen [11] found distinct energy metabolism patterns between continuous, high-intensity, and sprint interval training protocols, even when matched for mean intensity. These differences might be particularly relevant for tennis players who need to optimize both aerobic and anaerobic energy systems. The chronic effects of different training protocols have also been examined. Hebisz, Cortis [12] found that polarized training incorporating both HIIT and SIT led to significant improvements in aerobic fitness and cardiovascular health markers. However, the specific effects of SIT compared to HIIT on repeated sprint capacity in tennis players remain unclear.

Despite the growing body of research on high-intensity training methods, there is limited evidence comparing the effects of SIT and HIIT specifically on tennis performance parameters. Villafaina, Gimenez-Guervos Perez [13] demonstrated the safety and effectiveness of HIIT in rehabilitation settings, but the optimal protocol for improving tennis-specific performance remains debatable. Therefore, the purpose of this study is to compare the effects of SIT and HIIT on repeated sprint capacity and sport-specific performance in college-aged male tennis players. The hypothesis of this study was that SIT would lead to significantly greater improvements in repeated sprint ability, aerobic endurance (as measured by maximal completed level), and tennis-specific accuracy, compared to HIIT. It was further hypothesized that both training modalities would similarly enhance $VO_2$max and time to exhaustion, with no substantial differences in time to the onset of blood lactate accumulation and ventilatory threshold adaptations.

## Materials and methods

### Participants

A volunteer sample of 30 competitive collegiate tennis players participated in the study during the preparatory season. The study subjects were recruited between January 20 and January 25, 2025, and all data collection was completed prior to manuscript preparation. Although the sample size was limited to 15 participants per group due to the availability of competitive collegiate tennis players, a post-hoc power analysis was conducted using G*Power 3.1. Based on the observed large effect size for RSATT (partial $\eta^2 = 0.587$, equivalent to Cohen's $f = 1.19$), the statistical power $(1 - \beta)$ for detecting group × time interaction effects using a repeated-measures ANOVA (within–between interaction, $\alpha = 0.05$) was calculated to be $> 0.95$. This indicates that the study was adequately powered to detect meaningful training-related differences. Participants were randomly assigned to either the SIT or HIIT group using a computer-generated random sequence. To ensure allocation concealment, the randomization procedure was implemented by an independent researcher who was not involved in participant recruitment, assessment, or training supervision. Group assignments were revealed only after baseline testing had been completed. To ensure consistency in the characteristics of the participants and to accommodate the limited availability of female athletes from the tennis academy, the study focused exclusively on male tennis players. The participants engaged in 20 hours of training per week, with 3 hours allocated to technical and tactical tennis practice and 1 hour to physical conditioning based on traditional training. This was carried out on weekdays, and all participants were right-handed. In addition, all participants were required to meet the following inclusion criteria: they were required to be in good health and to have no severe injuries sustained during the six months preceding the study; furthermore, they were required to have a minimum of four years' experience of systematic tennis training. Prior to their participation in the study, all subjects were fully informed of the experimental procedures, potential benefits, and associated risks. Written informed consent was obtained from each subject prior to their participation. All tests were conducted at least 48 hours after a competitive match or strenuous training session. Subjects were required to participate in all training sessions, as well as the pre- and post-tests. The study was approved by the Research Ethics University of Chinese Academy of Sciences (Approval number: UCASSTEC25−004) and all procedures were conducted in accordance with the Declaration of Helsinki.

## Procedures

A longitudinal and randomized design was employed to investigate the effect of an 8-week court-based SIT intervention on the repeated sprint ability and performance parameters of tennis players. A two-group, repeated-measures design was employed, comprising a pre-test and a post-test. The participants were randomly allocated to the SIT and HIIT groups (SIT: n = 15, HIIT: n = 15) using stratified block randomization (Fig 1). Participants were randomly assigned to the SIT or HIIT groups using stratified block randomization. Stratification was based on baseline $VO_2max$ and age, ensuring that both aerobic fitness and maturity levels were balanced between groups. A block size of 4 was used to maintain equal group allocation across each stratum, with a 1:1 allocation ratio (SIT:HIIT). The randomization sequence was computer-generated using the RAND function in Microsoft Excel by an independent researcher not involved in the data collection or intervention delivery. The demographic data of the participants are presented in Table 1. Before the intervention, no significant differences were observed among the groups in terms of competition level, biometric training characteristics, anaerobic parameters, and anaerobic-specific performance. Furthermore, both groups maintained the same technical-tactical training agreed upon by the tennis academy during the intervention period. There were no reports of missed sessions or injuries. It should be noted that this study was this period.

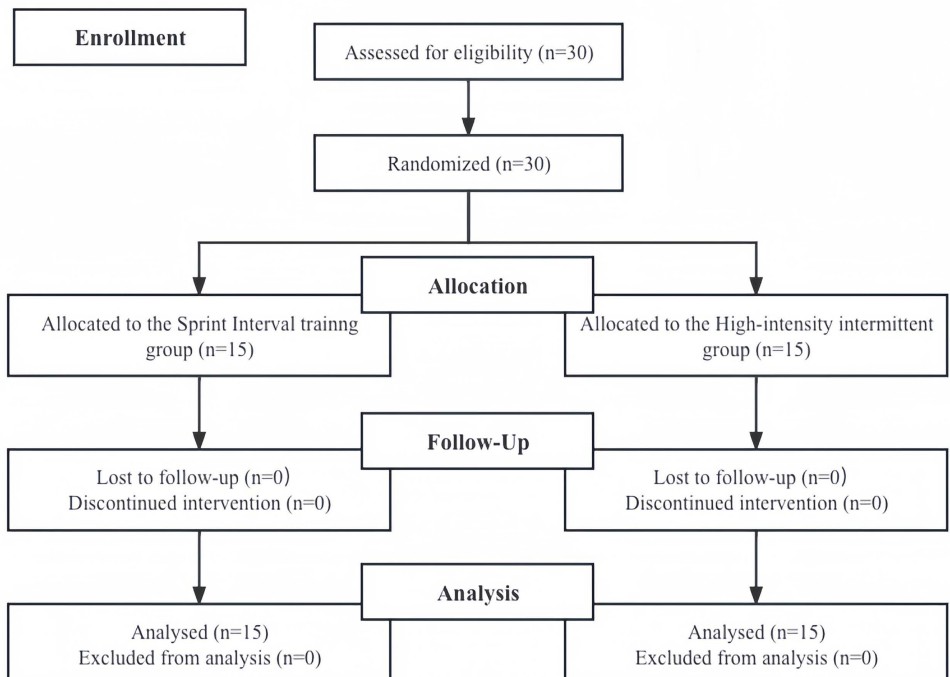

**Fig 1. Flow chart of the progress through the phases of the study according to the CONSORT statements.**

**Table 1. Physical characteristics of tennis players included in the analysis (baseline).**

|  | Age(year) | High(cm) | Body Mass(kg) | Training Background (year) | Competitive Level (ITN) |
|---|---|---|---|---|---|
| **SIT(n = 15)** | 22.36 ± 1.78 | 182.14 ± 4.33 | 75.72 ± 3.10 | 5.50 ± 0.94 | 4.54 ± 0.31 |
| **HIIT(n = 15)** | 22.43 ± 1.50 | 181.57 ± 4.88 | 75.95 ± 3.94 | 5.29 ± 1.14 | 4.64 ± 0.31 |

HIIT, high-intensity interval training as the control group; SIT, sprint interval training group; ITN, international tennis number.

Prior to the intervention, all participants were required to complete the tennis-specific repeated sprint ability (RSA) test, hit and turn tennis test (HTT) and the test to exhaustion specific to tennis (TEST). For each training and testing session, the participants engaged in a 10-minute standardized general warm-up and 10-minute cool-down protocol, comprising activities such as jogging, skipping, dynamic warm-up and stretching. The participants were required to refrain from engaging in any intensive exercise for a minimum of 48 hours prior to the commencement of the testing sessions. Furthermore, a minimum of 48 hours was to elapse between each session. The research was conducted during a preparatory training period that did not involve competition, and all measurements were undertaken in the morning, typically between 7:30 a.m. and 8:30 a.m.

## Measurements

Participants' physical performance was assessed using a testing battery performed in normoxia in a well-ventilated room at a constant temperature of ~21°C and ~55% relative humidity. Pre- and post-testing sessions were completed in the exact same sequence: (i) 20 min of standardized warm-up including athletic and acceleration drills; (ii), RSA test and HTT; and (iii) after 24 hours of rest, an incremental field test up to exhaustion [i. e., the so-called 'test to exhaustion specific to tennis' (TEST)]. Participants were asked to arrive at the testing sessions in a rested and hydrated state (at least 3 h after a meal and having avoided strenuous training in the preceding 24 h). Testing sessions for each participant were scheduled at the same time of day pre- and post-intervention, and conducted in the same indoor environment with a consistent order of assessments. All outcome assessments were conducted by trained testers who were blinded to the participants' group assignments to reduce the risk of bias.

### Hit and turn tennis test

The Hit and Turn Test was developed as an acoustically controlled progressive on-court fitness test for tennis players, which can be performed simultaneously by one or more players (Fig 2). The test involves specific movements along the baseline (i.e., side steps and running), combined with forehand and backhand stroke simulations at the doubles court corner (distance 11.0 m). At the beginning of each test level, the players stand with their racket in a frontal position in the middle of the baseline. Upon hearing a signal, the player turns sideways and runs to the prescribed (i.e., by the CD player) backhand or forehand corner. After making their shot, they return to the middle of the court using side steps or crossover steps (while looking at the net). When passing the middle of the baseline again, they turn sideways and continue to run to the opponent's opposite corner. The end of the test was considered when players fail to reach the cones in time or was no longer able to fulfill the specific movement pattern. Maximal completed level was used for the determination of the tennis-specific endurance capacity [14].

### Tennis-specific RSA test

To measure RSA, we used a test consisting of ten 20-m sprints departing every 20 s performed in a back and forth format [15] (i.e., 5 m + 10 m + 5 m) (Fig 3). This test was designed to measure tennis player's repeated sprint ability [16]. This format was designed to simulate tennis-specific movements and assess players' repeated sprint ability [16]. Players began each sprint from the center of the baseline, facing the net in a ready position. At the signal, they turned laterally and sprinted to the assigned backhand (left) or forehand (right) corner. They then executed a 180° turn at the cone (5 m), sprinted straight to the opposite corner (10 m), performed another 180° turn, and returned to the center (5 m). Timing was recorded using a photocell system (Sportronic TS01-R04, Germany), and three outcome variables were calculated: best sprint time ($RSA_{best}$), total sprint time ($RSA_{TT}$), and percentage decrement ($S_{dec}$), calculated as:

$$S_{dec} = 100 \times \left( \frac{Total\,sprint\,time}{Ideal\,sprint\,time \times Number\,of\,sprints} - 1 \right)$$

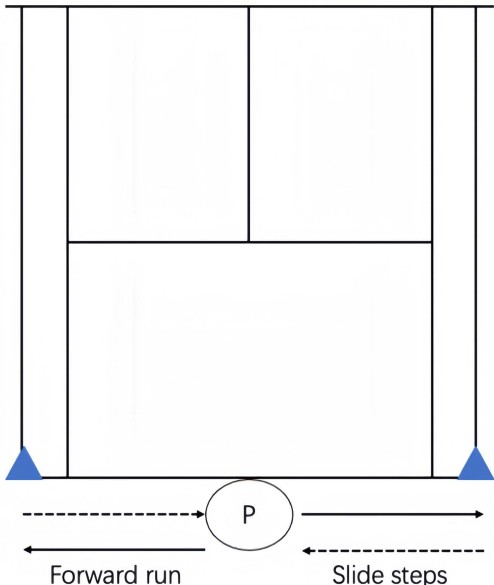

**Fig 2. Running and hitting court positions during the Hit and Turn Tennis Test.**

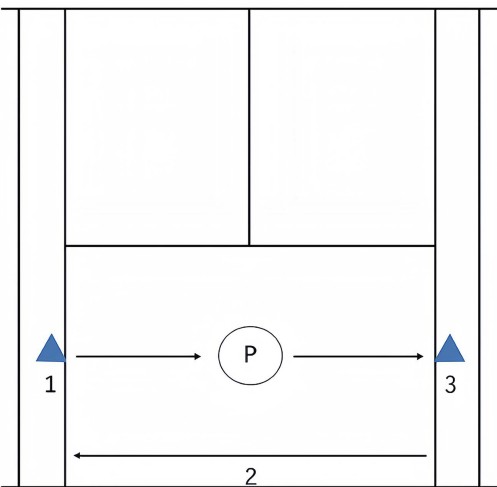

**Fig 3. Running pattern during the repeated-sprint ability (RSA) test.** •, light cells.

A preliminary shuttle sprint was conducted to determine each player's maximal single sprint time. If the first sprint in the RSA test was ≥ 2.5% slower than this reference value, the trial was stopped and repeated after a 5-minute rest.

### Test to exhaustion specific to tennis

Test to Exhaustion Specific to Tennis (TEST) was selected to assess participants' high-intensity intermittent performance(- Fig 3) [17], TEST is useful for combining physiological measurements with groundstroke performance to test a tennis player's endurance ability and athletic performance [18] (stroke accuracy and precision). Briefly, TEST consisted of hitting

balls thrown by a 'HighTOF' ball machine (Echouboulains, France) at constant velocity, alternating forehand and backhand strokes. The TEST started with a 2-min 'habituation'/warm-up phase where a ball frequency (BF) of 16 shots.min⁻¹ with balls thrown to the central area of the court (minimal lateral displacement) was adopted. After 1 min of passive rest (quiet standing), the first TEST stage began with a BF of 10 shots.min⁻¹, thereafter, increased by+2 shots.min⁻¹ every min until the stage corresponding to a BF of 22 shots.min⁻¹. Thereafter, the increment in BF was set at+1 shots.min⁻¹ until exhaustion. After each 1-min stage, a 30-s passive recovery break (quiet standing) was implemented.

Participants were required to perform 30 consecutive forehand smashes with maximum effort in a cross-court direction following a prescribed pattern. Slice strokes were not allowed, as they could alter the ball's trajectory and compromise the standardization of performance and physiological measurements. Participants were instructed to simulate match-like conditions and were told to "hit the ball with the best possible speed/accuracy ratio," similar to official competitions. Immediate feedback was provided to enhance engagement and motivation. A high-speed camera (240 fps) was used to capture shuttlecock landing points and ball velocity (BV). An accuracy target zone (1.5 m × 1.5 m) was marked in the deep backcourt. Hits were considered accurate if the shuttlecock landed fully within the target zone. In line with previous studies in elite players, any smash with a BV < 80 km/h or a BA < 50% (i.e., fewer than 15 accurate smashes out of 30) was considered unsuccessful. The final ball accuracy rate was calculated as the percentage of accurate smashes over the 30 trials.

TEST ended with participants' voluntary exhaustion or was stopped by the investigators if: (i) participants felt exhausted or failed to reach and hit the ball twice in a row, or (ii) they were no longer able to perform strokes with an acceptable execution technique and BV/BA declined, as determined by one experienced tennis national coach. Specifically, participants were given a warning the first time they disrespected the rules and were stopped on the second warning. TEST's performance was measured as total time to exhaustion (TTE).

## Evaluation of groundstroke performance

During TEST, groundstroke production was assessed via two main variables: BV and BA. BV (km.h⁻¹) was measured with the PlaySight®system (PlaySight Interactive, Ltd., Kokhav Ya'ir, Israel) which was approved by the International Tennis Federation (ITF) as a tennis player analysis technology for all ITF-sanctioned tournaments. For instance, correlations were reported between competitive level and BA (r = 0.61) [19] or stroke ratings (i. e., a surrogate of BA) (r = 0.94) [20]. BV and BA data were averaged for each TEST stage. Finally, because BV and BA better reflect the overall stroke precision in tennis when combined, the tennis performance (TP index) [18] was calculated as the product of these two variables. Consequently, if one factor was unchanged (e. g., BV) and the other one increased (e. g., BA), TP followed the pattern of BA. To determine the mean of each parameter measured over the entire TEST, calculation was based on the values obtained for each TEST stage. The PlaySight system has shown high validity and reliability in measuring ball placement and velocity in racket sports. The ICC for ball velocity was 0.96, and for ball accuracy was 0.92, indicating excellent test–retest reliability of the PlaySight system in our testing context.

## Physiological measurements

During TEST, expired air was analyzed (breath-by-breath measurements) for oxygen consumption ($\dot{V}O_2$) using a portable gas analyzer (Metamax II CPX system, Cortex®, Leipzig, Germany). Gas and volume calibration of the measurement device was performed before each test according to manufacturer's instructions. Heart rate (HR) was checked continuously (Suunto Ambit2®, Vantaa, Finland). Furthermore, 25 µL capillary blood samples were collected from fingertip and analyzed for blood lactate concentration (LT-1730; Arkray®, Kyoto, Japan) at the baseline, during TEST (i. e., during the 30-s recovery periods after every stage until a value of 4 mMol.L⁻¹ was obtained and thereafter every 2 stages), and 15 s after TEST exhaustion to assess maximal blood lactate concentration ($[La]_{max}$).

Detection of the second ventilatory threshold (VT2) was done by analyzing the points of change in slope (breaks in linearity) of ventilatory parameters [21,22]. VT2 was determined using the criteria of an increase in both $VE/\dot{V}O_2$ and VE/

$VCO_2$ [23]. VT2 assessment was made by visual inspection of graphs of time plotted against each relevant respiratory variable measured during testing. All visual inspections were carried out by two experienced exercise physiologists. The results were then compared and averaged. The difference in the individual determinations of VT2 was < 3%.

The onset of blood lactate accumulation (OBLA), defined as the exercise intensity corresponding to 4 $mMol.L^{-1}$ blood lactate concentration was also determined. By plotting each subject's blood lactate concentration against time of TEST completion and visually connecting the data points, we estimated the time to attain OBLA. This physiological variable has been shown to be a good predictor of endurance performance [24].

## Training program

The SIT and HIIT protocols were adapted from a previous study [1]. The training program took place during the preparatory period. Subjects trained 3 times per week for 8 weeks, with each session consisting of progressive adjustments in intensity, volume, and recovery periods to ensure a gradual overload. Specifically, each week the intensity was gradually increased by adjusting the duration of sprints and reducing recovery periods between intervals. For SIT, sprint duration was progressively increased from 15 to 30 seconds, and for HIIT, work-to-rest ratios were adjusted to maintain intensity at 90–95% HRmax. Both training interventions were performed on-court, separated by at least 48 hours. During each session, 3 intensive exercise bouts (i.e., HIIT or RST) were interrupted by an on-court tennis game (e.g., "2 against one player" [2:1 game]), where 2 players took turns in playing against the single one after each point. During each training session, each player played one time alone. The intensity of the 2:1 game was fixed between 75 and 85% of HRmax (Table 2). For the HIIT, the program consisted of replicating the movements executed during the Hit and Turn Test (i.e., side steps and running, combined with forehand and backhand stroke simulations; see Fig 1). The speed of running was controlled via beep signals from a CD, the stroke production being exactly simultaneous to the beep signals coming from it. The players performed 3 sets of 3 × 90-second runs at an intensity of the maximum level reached in the Hit and Turn Test. The intensity was individually adapted during the 8-week period considering the HR being above 90–95% of HRmax during the exercise (i.e., 270-second work, 540-second rest). Each set was separated by 180 seconds of active recovery, running around the court at an intensity of 70% of HRmax. For the RST, the program consisted of replicating the RSA shuttle test (Fig 2), performing 3 sets of 10 × 20-m shuttle sprints, separated by 20 seconds of passive recovery between repetitions (i.e., 50-second work, 150-second rest).

Before the commencement of the SIT or HIIT protocol and at the end of each run, a 5 µL blood sample was taken from the fingertip to determine the whole blood lactate concentration. Participants were verbally encouraged throughout both exercise protocols. All training sessions for both groups were supervised by an investigator with strength and conditioning experience. The Polar Team 2 System (Polar Electro Oy, Kemple, Finland) was used to monitor the heart rate of each player throughout each training session, with data later extracted from custom-specific software (Polar Team 2, Electro

**Table 2. Schematic representation of the training intervention.**

|  | Sprint interval training | High-intensity intermittent training |
|---|---|---|
| **Exercise** | sprints | movements of hit and turn test |
| **Intensity:** | all-out sprints | $V_{HTT}$ |
| **Volume:** | 5s/rep<br>10 reps/set,3 set | 90s/rep<br>3reps/set,3set |
| **Rest:** | 15sec/rep<br>8min/set (2:1 game:75–85%HRmax) | 180sec/rep (active recovery:70% of HRmax)<br>8min/set (2:1 game:75–85%HRmax) |
|  |  | The intensity was individually adapted during the 8-week period considering the HR being above 90–95% of HRmax during the exercise |

Sec: second; $V_{HTT}$: an intensity of the maximum level reached in the Hit and Turn Test (HTT).

Oy, Kemple, Finland), to obtain maximum heart rate (HRmax), time spent in each HRmax% zone and training impulse (TRIMP). TRIMP considers the training duration and intensity at the same time and reflects the comprehensive effect of training on the internal and external load of the athlete's body, as well as the load of medium and high-intensity training. The method to determine the athlete's TRIMP in the current study is based on the formula proposed by Edwards, a weight factor of each heart rate zone is given whereas the TRIMP per each zone is acquired by multiplying the exercise time [25]. Heart rate (HR) was continuously monitored during each training session using chest-strap HR monitors (Polar H10, Polar Electro Oy, Finland). The total time spent in five heart rate zones—Zone 1 (50–60% HRmax), Zone 2 (60–70%), Zone 3 (70–80%), Zone 4 (80–90%), and Zone 5 (90–100%)—was recorded and multiplied by zone-specific weighting factors (1–5, respectively). TRIMP was calculated as follows:

$$\textbf{TRIMP} = (\textbf{\textit{t}}_{\textbf{Z1}} \times \textbf{1}) + (\textbf{\textit{t}}_{\textbf{Z2}} \times \textbf{2}) + (\textbf{\textit{t}}_{\textbf{Z3}} \times \textbf{3}) + (\textbf{\textit{t}}_{\textbf{Z4}} \times \textbf{4}) + (\textbf{\textit{t}}_{\textbf{Z5}} \times \textbf{5})$$

The HRmax of each player was established using the peak value recorded by the monitoring system during the training. The total weekly TRIMP was calculated by summing the TRIMP values from all training sessions. Training protocols were periodically adjusted to maintain comparable weekly TRIMP values between groups, with variations controlled within ±10%. This approach ensured that both groups were exposed to similar overall cardiovascular loads, thereby allowing observed performance differences to be more confidently attributed to the type of training rather than discrepancies in training volume. Although the SIT and HIIT protocols inherently differed in work-to-rest ratios and intensity profiles, both were standardized in terms of session frequency (three sessions per week), session duration (~45 minutes), and heart rate–based internal load monitoring. Heart rate data were collected using Polar Team 2 monitors, and TRIMP values were used to quantify internal training load.

## Statistical analysis

Statistical analysis was performed using IBM SPSS software (version 26.0, Chicago, IL, USA). Data are presented as mean ± standard deviation (M ± SD). The Shapiro-Wilks test was used to verify normal distribution of all variables, and outliers (defined as studentized residuals exceeding 3 standard deviations from zero) were identified and removed. No outliers were detected or removed from the dataset. The effects of training interventions on physical fitness and tennis-specific performance parameters were analyzed using a two-way repeated-measures ANOVA (group × time). Dependent variables included: maximal completed level, RSATT, RSAbest, Sdec, total time to exhaustion (TTE), VO2max, HRmax, VEmax, [La]max, time to OBLA, time to VT2, ball velocity (BV), ball accuracy (BA), and tennis performance index (TP). The model factors were group (SIT and HIIT), time (pre and post), and their interaction (group × time). When significant interactions were detected, LSD post-hoc tests were conducted to identify specific differences. Additionally, within-group training effects (SIT or HIIT) were examined using separate one-way ANOVA models with time as the factor. Effect sizes were calculated using partial η² and interpreted as: small (<0.06), moderate (<0.14), or large (≥0.14) [26]. The significance level of these models was set at $p < 0.05$.

## Results

Following the 8-week intervention, a series of two-way repeated-measures ANOVAs revealed significant improvements in both groups across multiple outcome measures, with the SIT group demonstrating superior enhancements in repeated sprint ability and tennis-specific performance. For the maximal completed level in the Hit-and-Turn Tennis Test, a significant group × time interaction was observed ($F_{(1,28)} = 22.649$, $p < 0.05$, $\eta p^2 = 0.447$), with greater improvement in the SIT group (14.21 ± 1.05 to 18.21 ± 1.19) compared to the HIIT group (14.36 ± 0.84 to 16.50 ± 0.85). Similarly, TTE showed a significant interaction ($F_{(1,28)} = 12.350$, $p = 0.002$, $\eta p^2 = 0.306$), with the SIT group increasing from 588.43 ± 32.27 s to 670.14 ± 36.42 s, and the HIIT group from 596.64 ± 20.77 s to 641.71 ± 30.14 s. Both measures also demonstrated significant main effects of time ($p < 0.05$) (Table 3 and Fig 4).

**Table 3. Descriptive statistics of results for SIT and HIIT group before and after the 12-week training intervention.**

| | SIT | | | HIIT | | |
|---|---|---|---|---|---|---|
| | **Pre** | **Post** | **Partial $\eta^2$** | **Pre** | **Post** | **Partial $\eta^2$** |
| Maximal completed level | 14.21±1.05 | 18.21±1.19*# | 0.89 | 14.36±0.84 | 16.50±0.85* | 0.70 |
| $RSA_{TT}$(s) | 34.05±2.07 | 29.21±0.82*# | 0.89 | 33.45±1.83 | 31.58±1.86* | 0.55 |
| RSAbest(s) | 3.28±0.21 | 2.86±0.07*# | 0.82 | 3.22±0.18 | 3.03±0.18* | 0.50 |
| $S_{dec}$(%) | 3.96±1.68 | 2.25±1.28*# | 0.18 | 3.76±1.59 | 4.30±1.84 | 0.02 |
| Total time to exhaustion(TTE)(s) | 588.43±32.27 | 670.14±36.42*# | 0.83 | 596.64±20.77 | 641.71±30.14* | 0.59 |
| $VO_{2max}$(mL.min$^{-1}$.kg$^{-1}$) | 52.84±4.22 | 64.50±3.85*# | 0.64 | 51.79±5.15 | 59.6±4.44* | 0.45 |
| $HR_{max}$(beats.min$^{-1}$) | 195.29±5.17 | 191.14±4.00* | 0.22 | 193.86±4.45 | 192.43±3.65 | 0.03 |
| $VE_{max}$(L.min$^{-1}$) | 133.86±14.25 | 154.64±8.55*# | 0.54 | 136.86±11.03 | 146.29±7.85* | 0.19 |
| $[La]_{max}$(mMoles.L$^{-1}$) | 11.53±2.11 | 10.40±1.28 | 0.32 | 11.94±1.99 | 10.59±1.38* | 0.40 |
| Time to OBLA (s) | 345.00±44.77 | 478.07±23.03*# | 0.86 | 362.00±34.15 | 435.50±19.73* | 0.66 |
| Time to VT2 (s) | 371.79±36.53 | 470.36±29.61*# | 0.82 | 376.14±33.70 | 427.57±21.56* | 0.55 |
| Ball velocity (BV)(km/h) | 110.28±12.55 | 114.30±10.74 | 0.12 | 107.47±11.39 | 111.30±10.94 | 0.11 |
| Ball accuracy (BA)(%) | 57.29±8.97 | 70.14±5.33*# | 0.90 | 58.07±7.11 | 61.93±5.38* | 0.45 |
| tennis performance index (TP)(a.u.) | 66.36±4.70 | 75.50±5.02*# | 0.58 | 69.10±5.67 | 71.14±4.93 | 0.07 |

**Note**: RSATT, Repeated Sprint Ability Total Time; RSAbest, Best Sprint Time in Repeated Sprint Ability Test; Sdec,Sprint Decrement; TTE, Total Time to Exhaustion; VO$_2$max, Maximal Oxygen Uptake; HRmax, Maximal Heart Rate; VEmax, Maximal Ventilation; [La]max, Maximal Blood Lactate Concentration; OBLA, Onset of Blood Lactate Accumulation; VT2, Second Ventilatory Threshold; BV,Ball Velocity; BA,Ball Accuracy; TP, Tennis Performance Index; a.u.,Arbitrary Units.* Statistically significant difference between pre-and post-test, $p < 0.05$. # Statistically significant difference between group, $p < 0.05$.

Repeated sprint performance showed clear superiority of SIT. RSATT showed a significant interaction (F (1,28) = 39.740, p<0.01, ηp$^2$=0.587), with a greater reduction in the SIT group (34.05±2.07 s to 29.21±0.82 s) compared to the HIIT group (33.45±1.83 s to 31.58±1.86 s). A similar trend was observed in RSAbest (F (1,28) = 17.395, p<0.01, ηp$^2$=0.383), with improvements in both groups but a larger magnitude in SIT. Notably, Sdec decreased significantly in the SIT group (3.96±1.68% to 2.25±1.28%) but increased slightly in HIIT (3.76±1.59% to 4.30±1.84%), resulting in a significant interaction (F (1,28) = 4.908, p=0.036, ηp$^2$=0.149) and group effect (p=0.011) (Table 3 and Fig 4).

Physiological parameters exhibited distinct patterns. Although no interaction was found for VO$_2$max (F(1,28) = 2.541, p=0.123), both groups improved significantly over time (SIT: 52.84±4.22 to 64.50±3.85 mL·min$^{-1}$·kg$^{-1}$, ηp$^2$=0.644; HIIT: 51.79±5.15 to 59.60±4.44, ηp$^2$=0.450), with a main effect of group (F(1,28) = 6.398, p=0.018). VEmax showed a significant group×time interaction (F(1,28) = 4.546, p=0.043, ηp$^2$=0.140), with improvements in both groups, while HRmax demonstrated only a significant time effect (F(1,28) = 6.500, p=0.017), decreasing significantly in the SIT group. Blood lactate concentration ([La]max) decreased significantly in the HIIT group (11.94±1.99 to 10.59±1.38 mM), but not in SIT, although a significant main effect of time was observed (F(1,28) = 29.374, p<0.01) (Table 3 and Fig 4).

Metabolic threshold markers further highlighted the superiority of SIT. Time to OBLA showed a significant interaction (F (1,28) = 16.584, p<0.01, ηp$^2$=0.372), with SIT increasing from 345.00±44.77 s to 478.07±23.03 s, and HIIT from 362.00±34.15 s to 435.50±19.73 s. Time to VT$_2$ also demonstrated a significant interaction (F(1,28) = 13.546, p=0.01, ηp$^2$=0.326), favoring SIT (371.79±36.53 s to 470.36±29.61 s) over HIIT (376.14±33.70 s to 427.57±21.56 s) (Table 3).

Tennis-specific performance metrics showed that SIT induced greater improvements than HIIT. Although no interaction was observed for ball velocity (F(1,28) = 0.004, p=0.949), a main effect of time was present (p=0.014), with both groups showing minor improvements. However, ball accuracy improved significantly more in SIT (57.29±8.97% to 70.14±5.33%) compared to HIIT (58.07±7.11% to 61.93±5.38%), as evidenced by a significant interaction (F(1,28) = 13.811, p<0.001,

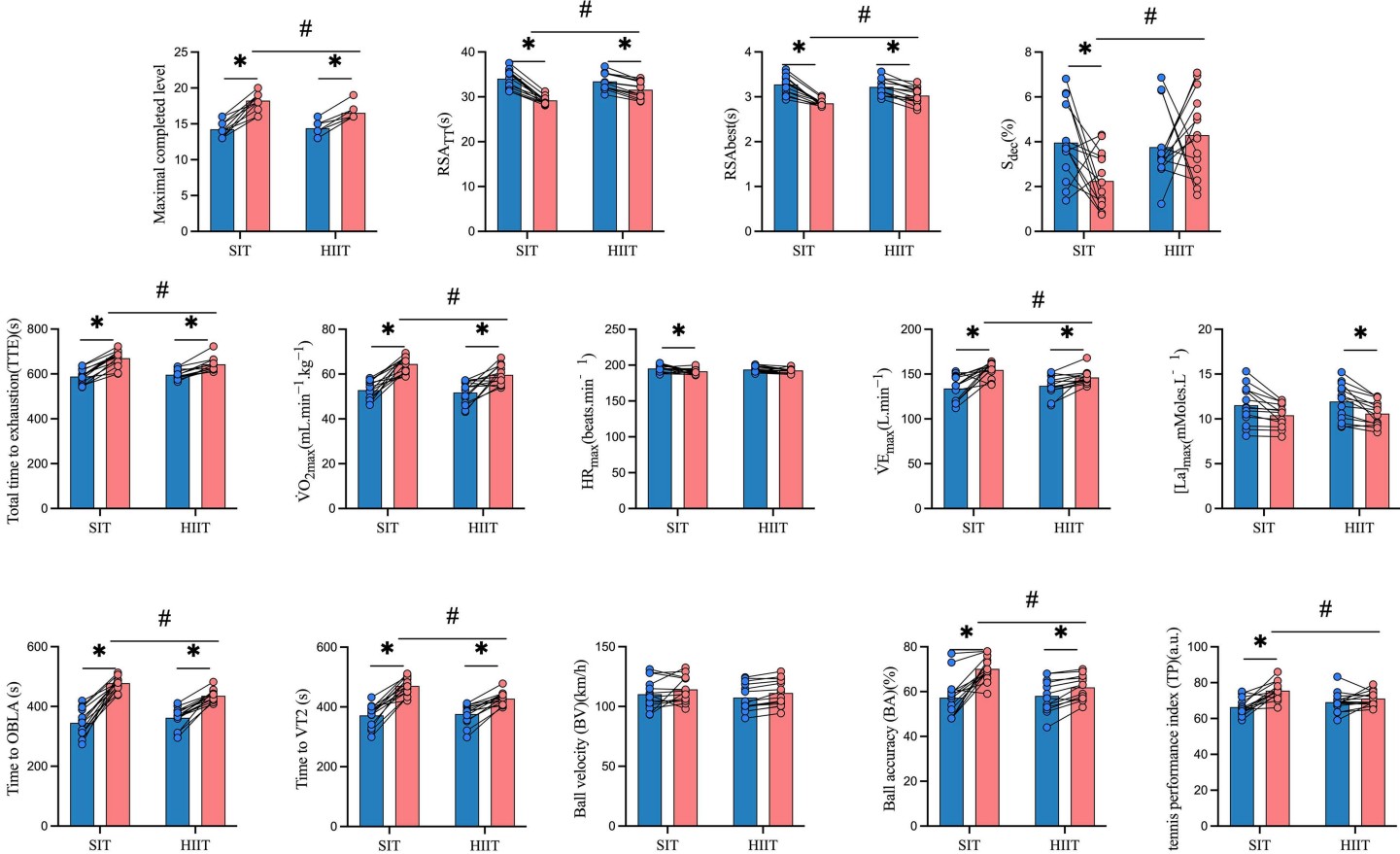

**Fig 4. Changes in repeated sprint ability and tennis-specific performance (RSATT, RSAbest, and Sdec) before and after the 8-week training intervention in the SIT and HIIT groups.** Abbreviations: RSATT = Repeated Sprint Ability Total Time; Sdec = Sprint Decrement; $VO_2$max = Maximal Oxygen Uptake; OBLA = Onset of Blood Lactate Accumulation; TP = Tennis Performance Index; BA = Ball Accuracy. * Statistically significant difference between pre-and post-test, $p < 0.05$. # Statistically significant difference between group, $p < 0.05$.

$\eta p^2 = 0.331$). TP also showed a significant interaction ($F(1,28) = 10.936$, $p = 0.003$, $\eta p^2 = 0.281$), with SIT demonstrating superior improvements ($66.36 \pm 4.70$ to $75.50 \pm 5.02$) versus HIIT ($69.10 \pm 5.67$ to $71.14 \pm 4.93$) (Table 3 and Fig 4).

## Discussion

The primary objective of this study was to compare the effects of SIT and HIIT on repeated sprint capacity and sport-specific performance in college-aged male tennis players. The results demonstrated that SIT was more effective in improving several aspects of repeated sprint performance and tennis-specific measures, including the maximal completed level of the hit-and-turn tennis test, RSAbest, RSATT, Sdec, TTE during tennis-specific endurance testing, $VO_2$max, and VE max. Furthermore, SIT significantly improved the time to OBLA, time to VT2, tennis performance index, and ball accuracy, while also lowering HR max and [La] max during tests to exhaustion specific to tennis. HIIT, on the other hand, showed improvements in several variables, including the maximal completed level of the hit-and-turn tennis test, RSAbest, RSATT, TTE, $VO_2$max, VE, OBLA, and VT, but did not affect tennis-specific performance metrics such as ball accuracy or the tennis performance index.

SIT demonstrated effects on repeated sprint capacity in college-aged male tennis players, as indicated by significant improvements in RSAbest, RSATT, and Sdec. These outcomes suggest that SIT may better enhance the ability to perform

multiple, high-intensity efforts with minimal decrement in performance, which is crucial for tennis players who frequently engage in repeated sprints during matches [27]. This study's findings are consistent with prior research demonstrating the effectiveness of SIT in enhancing repeated sprint ability and high-intensity performance. For instance, Buchheit and Laursen [28] emphasized the effects of SIT on improving sprint capacity, which aligns with our results showing significant improvements in RSAbest, RSATT, and Sdec. Additionally, the significant reductions in Sdec suggest enhanced fatigue resistance during repeated high-intensity activities, a critical factor in tennis matches [29]. These effects are likely mediated by a range of SIT-induced physiological adaptations. First, the maximal sprint nature of SIT promotes greater recruitment of type II muscle fibers, which are crucial for explosive efforts such as repeated sprinting and directional changes seen in tennis. This neuromuscular adaptation enhances the rate of force development, contributing to improvements in RSAbest and RSATT. Second, SIT enhances mitochondrial biogenesis and increases oxidative enzyme activity (e.g., citrate synthase, succinate dehydrogenase), which support improved aerobic energy production during recovery phases between sprints. Third, SIT has been shown to upregulate monocarboxylate transporters (MCT1 and MCT4), facilitating more efficient lactate transport and clearance, thereby delaying fatigue as evidenced by the reduced Sdec and lower [La] max observed in our study. Together, these adaptations enable athletes to maintain sprint performance across repeated efforts while preserving technical execution during prolonged match-like conditions [30–32]. In contrast, the significant reduction in [La]max observed only in the HIIT group may be explained by the moderate-to-high intensity but longer duration work intervals, which are known to stimulate oxidative adaptations. These include increased mitochondrial density and upregulation of monocarboxylate transporters (e.g., MCT1), which facilitate more effective lactate clearance. This suggests that HIIT may enhance metabolic efficiency during prolonged submaximal efforts. However, the absence of concurrent improvements in tennis-specific technical outcomes (e.g., ball accuracy or TP index) indicates that reduced blood lactate accumulation alone may not directly translate into sport-specific performance gains. Rather, it may reflect a general improvement in recovery capacity during extended intermittent play. However, unlike previous studies, our research extended this evidence to sport-specific contexts by incorporating tennis-specific skill metrics [33,34]. This highlights SIT's unique ability to not only improve physical performance but also enhance specific performance for tennis players [35]. The significant improvements in specific performance as reflected by the tennis performance index and ball accuracy, coupled with the reduced HR max and [La] max, underscore SIT's potential in enhancing physical performance and maintaining technical stability in tennis [18,36]. SIT's effectiveness in reducing lactate accumulation and extending the time to exhaustion suggests enhanced aerobic and anaerobic fitness, which may contribute to better match endurance and recovery.

In contrast, while HIIT did improve certain physiological measures such as VO$_2$max, VE, OBLA, and VT, it did not lead to significant improvements in the tennis-specific metrics (i.e., tennis performance index, ball accuracy), nor did it reduce HR max or [La] max as effectively as SIT. These results are consistent with findings by Milanović, Sporiš [37], and recent evidence from Morais, Kilit [38], who observed improvements in aerobic capacity and speed following HIIT-based tennis training. However, our study revealed HIIT's limited impact on sport-specific measures, such as ball accuracy and the tennis performance index, supporting observations by Iaia and Bangsbo [39] that HIIT's adaptations may not fully translate to sports requiring high-intensity intermittent performance with technical precision. This discrepancy may stem from the different training modalities employed by SIT and HIIT. SIT, characterized by short, maximal effort sprints interspersed with recovery, is particularly well-suited for improving explosive power, sprint capacity, and sport-specific skill, which are key components of tennis performance. HIIT, with its longer, submaximal efforts, may be more beneficial for improving overall cardiovascular fitness, but may not optimally target the rapid recovery and explosive energy demands required in tennis.

The effects of SIT on repeated sprint ability and tennis-specific performance can be attributed to several underlying physiological mechanisms. First, SIT's maximal intensity and short recovery intervals result in greater recruitment of type II muscle fibers, which are critical for high-intensity and explosive movements such as sprints and changes of direction [28,40,41]. This neuromuscular adaptation enhances the capacity to generate force rapidly, contributing to the observed improvements in RSAbest and RSATT, as well as the reduced sprint decrement (Sdec), indicating improved fatigue

resistance across repeated efforts [42,43]. Second, the improvements in time to exhaustion (TTE) and reductions in [La] max can be attributed to SIT's enhancement of lactate clearance and buffering capacity. SIT stimulates metabolic adaptations such as upregulation of monocarboxylate transporters (MCT1 and MCT4), which facilitate intramuscular lactate shuttling and removal from the bloodstream [44,45]. These changes are associated with delayed onset of blood lactate accumulation (OBLA) and lower lactate accumulation during tennis-specific endurance efforts [46]. Third, SIT promotes mitochondrial biogenesis and elevates oxidative enzyme activity (e.g., citrate synthase, succinate dehydrogenase), which improve oxidative energy production and reduce reliance on anaerobic glycolysis during sustained efforts [47,48]. These mitochondrial adaptations contribute to greater ATP availability, prolonged endurance capacity, and reduced metabolic acidosis under repeated sprint conditions. Finally, the reductions in HR max and improvements in ventilatory efficiency (VE) observed in SIT participants likely reflect enhanced autonomic regulation and cardiovascular function. SIT has been associated with increased stroke volume, improved cardiac output, and greater vagal tone, which collectively reduce heart rate responses at submaximal and maximal workloads [32,49–51]. These adaptations are especially beneficial in intermittent sports like tennis, where efficient cardiovascular recovery between high-intensity efforts is essential.

The findings of this study offer meaningful insights for tennis coaches and physical conditioning professionals. Given the superior improvements observed in repeated sprint ability, aerobic capacity, and stroke performance, coaches may consider incorporating SIT protocols into their regular training routines. SIT sessions, characterized by short-duration maximal efforts with brief recoveries, can be adapted to mimic the intermittent high-intensity nature of tennis matches. Moreover, these sessions are time-efficient and can be implemented 2–3 times per week alongside technical and tactical training. For adolescent athletes, SIT may serve as a practical and effective method to improve both physiological conditioning and match-related skills within a limited training schedule.

Despite its strengths, this study has several limitations that should be acknowledged. First, the lack of a control group limits the ability to definitively attribute observed performance improvements solely to the SIT and HIIT interventions. Future studies should include a control group to account for potential confounding factors and natural performance variations over time. Second, potential learning effects in the testing protocols may have influenced the results, particularly for the tennis-specific performance assessments (e.g., tennis performance index, ball accuracy). Although familiarization sessions were provided, repeated exposure to the tests may have led to improvements independent of training-induced physiological adaptations. Third, the sample was limited to a homogeneous group of college-aged male tennis players with similar training backgrounds and competitive levels. This restricts the generalizability of the findings to other populations, such as female players, youth or veteran athletes, and individuals of varying skill levels or training histories. Additionally, the relatively short duration of the intervention precludes conclusions about the long-term sustainability and potential risks (e.g., injury incidence) of these training methods. Finally, although tennis-specific performance metrics were included, important aspects such as tactical decision-making, psychological resilience, and in-game performance were not assessed. Validated tennis-specific testing protocols such as the TSIAT could further enhance methodological rigor [52]. Future research should address these limitations to provide a more comprehensive understanding of how SIT and HIIT influence tennis performance in ecologically valid settings.

From a practical standpoint, these results suggest that incorporating SIT sessions 2–3 times per week during the preparatory phase of a training cycle may optimize improvements in repeated sprint capacity, aerobic power, and tennis-specific performance. A typical SIT session could involve 6–8 repetitions of maximal 20–30 meter sprints, each separated by 20–30 seconds of passive or low-intensity active recovery (e.g., walking). These sessions can be performed on separate conditioning days or placed after technical-tactical drills to simulate match-specific fatigue conditions. For adolescent or time-constrained athletes, this format offers a highly time-efficient method to develop both anaerobic and aerobic systems without significantly increasing overall training volume. Coaches are advised to adjust SIT protocols based on the athlete's training status, phase of the season, and recovery capacity. For example, during early pre-season phases, lower sprint volumes (e.g., 4–6 sprints) may be appropriate, while peak training phases can include higher sprint volumes

or shorter recovery intervals to increase training density. Internal load monitoring tools such as RPE (Rating of Perceived Exertion), heart rate variability (HRV), or session duration-adjusted TRIMP (Training Impulse) scores may be used to individualize progression and recovery. Importantly, although the SIT protocol in this study involved straight-line sprints, incorporating tennis-specific movement patterns (e.g., court-based agility drills or directional changes) may further enhance transfer to on-court performance, particularly during the tapering phase before competition. In contrast, HIIT protocols (e.g., 4×4 minute runs at 90–95% HRmax with 3-minute rest) may be more suitable during general preparation phases to build cardiovascular endurance and $VO_2$max. A periodized approach that integrates HIIT early in the macrocycle and gradually transitions to SIT in later stages may maximize both general conditioning and sport-specific performance outcomes.

Several limitations of this study should be acknowledged. First, although assessors were blinded to group assignment, participants and coaches were not, which may have introduced performance bias. Second, the sample consisted exclusively of male adolescent tennis players with similar baseline fitness levels, limiting the generalizability of our findings to female athletes, players from other racket sports, or those with different training backgrounds. Third, the intervention lasted eight weeks; while this duration is sufficient for observing short-term adaptations, it may not capture long-term training effects or sustainability. Finally, no follow-up was conducted to assess retention or delayed effects after the training period. Future research should aim to include more diverse participant samples, consider longer-term protocols, and evaluate post-intervention retention to improve the external validity of the findings. Additionally, although both training protocols were standardized in terms of frequency, duration, and internal load monitoring (e.g., TRIMP, HR zones), we did not precisely match the external mechanical load (e.g., total sprint distance, movement repetitions) or directly quantify metabolic stress (e.g., EPOC, hormonal responses). These factors may have contributed to the differences in physiological adaptations and performance outcomes. As SIT and HIIT inherently impose different neuromuscular and metabolic demands, we acknowledge that the observed effects could be influenced by both the nature of the training stimuli and the unequal distribution of cumulative load. Future studies should consider matching training load more rigorously or use objective external load metrics and post-exercise physiological markers to further isolate the specific effects of training modality.

## Conclusion

This study confirms that Sprint Interval Training (SIT) outperforms High-Intensity Interval Training (HIIT) in enhancing repeated sprint capacity and maintaining technical stability in tennis-specific performance. SIT demonstrated greater improvements in RSATT, Sdec, and the tennis performance index, alongside significant gains in aerobic fitness and fatigue resistance. These findings highlight SIT as an effective, sport-specific training strategy for tennis players and similar high-intensity intermittent sports.

## Supporting information

**S1 File. Data.**
(XLSX)

## Author contributions

**Conceptualization:** Jing Fan, Ke Sun, Xu Liu, Tianyu Zhu.

**Data curation:** Jing Fan, Yue Li.

**Project administration:** Tianyu Zhu.

**Software:** Xu Liu, Yue Li.

**Writing – original draft:** Jing Fan, Ke Sun, Xu Liu, Tianyu Zhu.

**Writing – review & editing:** Jing Fan, Xu Liu.

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
