## [Decision Letter · Decision Letter 0]

19 May 2025

PONE-D-25-21436Effects of sprint interval training compared to high intensity interval training on repeated sprint capacity and sport-specific performance in college-aged male tennis playersPLOS ONE

Dear Dr. Sun,

Thank you for submitting your manuscript to PLOS ONE. After careful consideration, we feel that it has merit but does not fully meet PLOS ONE’s publication criteria as it currently stands. Therefore, we invite you to submit a revised version of the manuscript that addresses the points raised during the review process.

We look forward to receiving your revised manuscript.

Kind regards,

Fenghua Sun

Academic Editor

PLOS ONE

3. We note that there is identifying data in the Supporting Information file <file name>. Due to the inclusion of these potentially identifying data, we have removed this file from your file inventory. Prior to sharing human research participant data, authors should consult with an ethics committee to ensure data are shared in accordance with participant consent and all applicable local laws.

-Location data

Reviewers' comments:

Reviewer's Responses to Questions

**Comments to the Author**

1. Is the manuscript technically sound, and do the data support the conclusions?

Reviewer #1: Yes

Reviewer #2: Yes

2. Has the statistical analysis been performed appropriately and rigorously? 

Reviewer #1: Yes

Reviewer #2: Yes

3. Have the authors made all data underlying the findings in their manuscript fully available?

Reviewer #1: Yes

Reviewer #2: No

4. Is the manuscript presented in an intelligible fashion and written in standard English?

Reviewer #1: Yes

Reviewer #2: Yes

5. Review Comments to the Author

Reviewer #1: General Comments

The manuscript investigates the effects of sprint interval training (SIT) compared to high-intensity interval training (HIIT) on tennis-specific performance in collegiate male tennis players. This research addresses a relevant topic in sports science, as optimizing training protocols for tennis players can enhance performance and potentially reduce injury risk.

Major Weaknesses:

1. Intervention Duration Inconsistency: The study describes both a "six-week" (line 107) and "8-week" intervention (lines 19, 233), creating confusion about the actual duration of the study.

2. Methodological Clarity Issues: The SIT and HIIT protocols lack precise descriptions of progression throughout the intervention period. Additionally, the manuscript suggests that training was conducted "during the off-season" (line 117) but also "during the preparatory season" (line 90), which is contradictory.

3. Statistical Reporting Inconsistencies: Some statistical results are reported with varying formats and levels of detail throughout the manuscript. The reporting of effect sizes is inconsistent, and some p-values do not match across text and tables.

4. Lack of Blinding Procedures: There is no mention of any blinding procedures for testers during pre- and post-intervention assessments, which could have introduced bias.

5. Limited Discussion of Physiological Mechanisms: While the authors attempt to explain the physiological mechanisms behind the observed differences between SIT and HIIT, these explanations lack sufficient references to supporting literature.

Minor Weaknesses:

1. Several grammatical and typographical errors throughout the manuscript.

2. Figures lack clear legends and comprehensive captions.

3. Incomplete discussion of limitations and generalizability of findings.

4. Table 3 presents inconsistent formatting of statistical significance markers.

5. Limited discussion of practical applications for tennis coaches and practitioners.

Specific Comments

Line 18-19: The study is described as involving 30 tennis players for an "8-week intervention," but later in line 107, it's referred to as a "six-week court-based SIT intervention." Clarify the actual duration of the intervention.

Line 90-91: The study claims recruitment was from "January 20, 2025, to January 25, 2025" which is a future date. Please correct.

Line 90-95: The manuscript states the study was conducted "during the preparatory season," but later (line 117) mentions it was "during the off-season." Clarify which is correct.

Line 108: The phrase "a longitudinal and randomized controlled experimental design" is redundant since randomized controlled already implies experimental design.

Line 111: More details on the stratified block randomization procedure are needed, including what factors were used for stratification.

Line 135-136: The Hit and Turn Test description refers to "Figure 1," but this figure appears to be the CONSORT flow diagram, not the test setup. This creates confusion.

Line 149-154: The RSA test description references "Figure 2," but the description doesn't fully match what is shown in the figure. Improve the clarity of the figure or description.

Line 184-190: The description of the TEST protocol lacks clarity on how ball accuracy was determined. Provide specific criteria.

Line 198-205: The methods for measuring ball velocity and accuracy need more detailed description, particularly the validity and reliability of the PlaySight system for these measurements.

Line 229-230: The statement that "The training program took place during the preseason period" contradicts earlier mentions of "preparatory season" and "off-season."

Line 251-262: More details are needed on how TRIMP was calculated and used to ensure equal training loads between groups.

Line 271: The statement about "outliers (defined as studentized residuals exceeding 3 standard deviations from zero) were identified and removed" should specify how many outliers were removed and from which variables.

Line 279-284: The results reporting for maximal completed level shows inconsistent formatting of statistics compared to other results.

Line 310-312: The paper reports that "Blood lactate concentration showed significant reduction only in HIIT group" but doesn't explain why this might have occurred or its practical significance.

Line 339-346: This paragraph makes claims about SIT's effectiveness on repeated sprint capacity without sufficient citations to support the mechanisms proposed.

Line 371-390: The explanation of physiological mechanisms lacks sufficient references, particularly when discussing muscle fiber recruitment, lactate clearance, and autonomic regulation.

Line 392-399: The limitations section fails to discuss important limitations such as lack of a control group, potential learning effects in the testing protocols, and the homogeneous sample.

Lines 414-506: The manuscript would benefit from incorporating more recent and relevant literature related to sport-specific testing, training load quantification, and interval training methodologies. I recommend the authors consider including the following references to strengthen their literature framework and discussion of mechanisms:

1. Dhahbi, W., Hachana, Y., Souaifi, M., Souidi, S., & Attia, A. (2024). Tennis-specific incremental aerobic test (TSIAT): construct validity, inter session reliability and sensitivity. Tunisian Journal of Sports Science and Medicine, 2(1), 25-32. This paper provides validated tennis-specific testing protocols that would enhance your methodological approach and interpretation of sports-specific adaptations.

2. Dhahbi, W., Chaabene, H., Pyne, D. B., & Chamari, K. (2024). Standardizing the quantification of external load across different training modalities: a critical need in sport-science research. International Journal of Sports Physiology and Performance, 19(11), 1173-1175. This work addresses the critical issue of training load standardization across different modalities like SIT and HIIT, which would strengthen your methodology section and discussion of comparative effects.

3. Turki, O., Dhahbi, W., Padulo, J., Khalifa, R., Ridène, S., Alamri, K., ... & Chamari, K. (2019). Warm-up with dynamic stretching: positive effects on match-measured change of direction performance in young elite volleyball players. International Journal of Sports Physiology and Performance, 15(4), 528-533. This research examines performance enhancement in an intermittent sport with change-of-direction demands similar to tennis, offering parallels to your study's assessment of repeated sprint ability.

4. Farhani, Z., Ghouili, H., Dhahbi, W., Ammar, A., Ben Aissa, M., Bouzouraa, M. M., ... & Ben Ezzeddine, L. (2025). Optimizing the number of players and training bout durations in soccer small‐sided games: Effects on mood balance and technical performance. European Journal of Sport Science, 25(3), e12208. This recent work explores technical performance preservation during high-intensity intermittent exercise, which aligns with your findings on technical stability during tennis-specific performance.

5. Dhahbi, W., Briki, W., Heissel, A., Schega, L., Dergaa, I., Guelmami, N., et al. (2025). Physical Activity to Counter Age-Related Cognitive Decline: Benefits of Aerobic, Resistance, and Combined Training - A Narrative Review. Sports Medicine (Open), In press. This forthcoming review discusses the differential physiological adaptations to various training modalities including combined approaches, which would enhance your discussion of the mechanisms underlying the superior effects of SIT observed in your study. Incorporating these recent references would strengthen the theoretical framework, methodological justification, and interpretation of your findings, particularly in relation to sport-specific adaptations and the physiological mechanisms underlying the observed training effects.

Table 3: The formatting of the table is inconsistent, with some cells containing multiple decimal places while others have different precision. Standardize the presentation.

Figure 4: The figure showing pre-post changes is difficult to interpret without better labeling and a more detailed caption explaining the statistical significance markers.

Reviewer #2: Title: Effects of sprint interval training compared to high intensity interval training on repeated

sprint capacity and sport-specific performance in college-aged male tennis players

This study compared the effects of sprint interval training (SIT) and high-intensity interval training (HIIT) on repeated sprint capacity and tennis-specific performance in male college tennis players. After an 8-week intervention, the SIT group showed more significant improvements in tennis-specific performance (ball accuracy, tennis performance index), repeated sprint ability (RSATT, RSAbest, Sdec), and aerobic capacity (VO₂max) compared to the HIIT group. Although both training methods increased aerobic fitness, it was concluded that SIT is a more efficient strategy for high-intensity intermittent sports such as tennis.

Thank you for the interesting insights your article provides. The manuscript is well-written and deals with an important topic. I have thoroughly reviewed the manuscript and would like to make some suggestions for your review.

"Line 30: The abbreviations 'OBLA' and 'VT2' should be written in their long forms at first use, followed by the abbreviations in parentheses. For example: 'the onset of blood lactate accumulation (OBLA) and second ventilatory threshold (VT2)'."

Line 88: The sample size is small (n = 15 per group), and no power analysis is provided to support the adequacy of the design for detecting meaningful effects.

Line 111: selection of subjects

The manuscript mentions "stratified block randomization" but fails to specify:

Stratification criteria (e.g., baseline VO₂max, tennis experience, or performance metrics).

Block size (e.g., 4 or 6 participants per block) and allocation ratio (e.g., 1:1 for SIT vs. HIIT).

Method of sequence generation (computerized tool? manual randomization?).

Without these details, the reproducibility of the study is compromised. please add the required explanation

Line 230: "Further clarification on the standardization of training protocols is needed.

Load Intensity Comparison: Although HIIT and SIT protocols used different work-to-rest ratios, the total workload (e.g., mechanical load) and metabolic stress (e.g., EPOC) were not matched between conditions. This makes it unclear whether the observed adaptations resulted from the training type itself or from unequal load distribution.

Line 238: While the HIIT protocol used tennis-specific movements, the SIT used general sprints. This leaves it unclear whether SIT's superiority in technical skills is due to training specificity or a genuine physiological advantage. This section requires further clarification.

Line 327: Enhance the discussion section using references.

- Morais, J. E., Kilit, B., Arslan, E., Bragada, J. A., Soylu, Y., & Marinho, D. A. (2024). Effects of On-Court Tennis Training Combined with HIIT versus RST on Aerobic Capacity, Speed, Agility, Jumping Ability, and Internal Loads in Young Tennis Players. Journal of Human Kinetics, 95, 173.

- Fernandez-Fernandez, J., Sanz-Rivas, D., Kovacs, M. S., & Moya, M. (2015). In-season effect of a combined repeated sprint and explosive strength training program on elite junior tennis players. The Journal of Strength & Conditioning Research, 29(2), 351-357.

- Brechbuhl, C., Schmitt, L., Millet, G. P., & Brocherie, F. (2018). Shock microcycle of repeated-sprint training in hypoxia and tennis performance: Case study in a rookie professional player. International Journal of Sports Science & Coaching, 13(5), 723-728.

Table 3: should list the abbreviated terms in the first column, with their full explanations provided at the end of the table.

Line 414: the reference section needs to be checked

References 1 and 35 are the same

References 19-22 are the same

References 19-28 needs to be checked

6. PLOS authors have the option to publish the peer review history of their article (what does this mean? ). If published, this will include your full peer review and any attached files.

**Do you want your identity to be public for this peer review?** For information about this choice, including consent withdrawal, please see our Privacy Policy .

Reviewer #1: **Yes: ** Wissem Dhahbi

Reviewer #2: No

---

## [Author Response · Author response to Decision Letter 1]

24 Jul 2025

Dear Editor and Reviewers,

We sincerely thank you for the time and effort you have invested in reviewing our manuscript entitled “Effects of sprint interval training compared to high intensity interval training on repeated sprint capacity and sport-specific performance in college-aged male tennis players.” We highly appreciate the constructive and detailed comments, which have significantly improved the quality and clarity of our work.

We have carefully addressed all the concerns raised by both reviewers and have revised the manuscript accordingly. A detailed point-by-point response has been prepared and is included below, with specific modifications made in the manuscript highlighted accordingly.

---

## [Decision Letter · Decision Letter 1]

13 Aug 2025

PONE-D-25-21436R1Effects of sprint interval training compared to high intensity interval training on repeated sprint capacity and sport-specific performance in college-aged male tennis playersPLOS ONE

Dear Dr. Sun,

Thank you for submitting your manuscript to PLOS ONE. After careful consideration, we feel that it has merit but does not fully meet PLOS ONE’s publication criteria as it currently stands. Therefore, we invite you to submit a revised version of the manuscript that addresses the points raised during the review process.

We look forward to receiving your revised manuscript.

Kind regards,

Fenghua Sun

Academic Editor

PLOS ONE

Journal Requirements:

**Additional Editor Comments:**

Please check the format of references, e.g., should be consistent in using "()" or "[]" to indicate reference no.

Reviewers' comments:

Reviewer's Responses to Questions

**Comments to the Author**

1. If the authors have adequately addressed your comments raised in a previous round of review and you feel that this manuscript is now acceptable for publication, you may indicate that here to bypass the “Comments to the Author” section, enter your conflict of interest statement in the “Confidential to Editor” section, and submit your "Accept" recommendation.

Reviewer #1: All comments have been addressed

2. Is the manuscript technically sound, and do the data support the conclusions?

Reviewer #1: Yes

3. Has the statistical analysis been performed appropriately and rigorously? 

Reviewer #1: Yes

4. Have the authors made all data underlying the findings in their manuscript fully available?

Reviewer #1: Yes

5. Is the manuscript presented in an intelligible fashion and written in standard English?

Reviewer #1: Yes

6. Review Comments to the Author

Reviewer #1: General Comments

The authors have made substantial improvements to the manuscript following the previous review. The study design comparing sprint interval training (SIT) to high-intensity interval training (HIIT) in tennis players addresses an important research question with practical applications. The methodology is generally sound, and the statistical analyses are appropriate. However, several minor issues require attention before the manuscript can be accepted for publication.

Specific Comments

Abstract: Page 1, Line 19: The phrase "8-week intervention" should be clarified as "8-week training intervention" to specify the nature of the intervention more precisely.

Page 2, Line 25: The statistical reporting format is inconsistent. Consider standardizing the presentation of F-values, p-values, and effect sizes throughout the abstract.

Introduction: Page 3, Line 65: The sentence beginning "This suggests that both HIIT and SIT might be beneficial..." creates an awkward transition. Consider restructuring this paragraph for better flow.

Page 4, Line 85-88: The hypothesis statement could be more specific about the expected magnitude of differences between training modalities.

Methods: Page 5, Line 92-93: The statement "Recruitment and data collection were completed between January and March 2025, prior to manuscript preparation" contains a future date that appears to be an error. This needs immediate correction.

Page 5, Line 99: Similarly, "The study subjects' recruitment period was from January 20, 2025, to January 25, 2025" uses future dates that require correction.

Page 6, Line 131: The phrase "conducted during the preparatory season" appears twice in close proximity. Consider removing the redundancy.

Page 8, Line 176: The calculation formula for Sdec should be presented more clearly, perhaps as a separate equation rather than embedded in the text.

Results: Page 15, Line 319: The F-value presentation format should be consistent throughout. Some instances use F(1,28) while others omit the degrees of freedom format.

Page 16, Line 343: The phrase "435�50±19.73*" contains what appears to be a formatting error with the comma placement.

Discussion: Page 18, Line 375: The sentence "These effects are likely mediated by SIT-induced physiological adaptations..." could benefit from more specific mechanistic details.

Page 21, Line 461: The practical recommendations section would benefit from more specific implementation guidelines for coaches.

Tables and Figures: Table 3, Page 33: The formatting of statistical significance markers could be more consistent. Some entries use asterisks while others use different notation systems.

Figure 4, Page 35: The figure caption could be more descriptive about what specific changes are being highlighted.

References: Page 24-31: Several references contain formatting inconsistencies, particularly in journal name abbreviations and page number presentations.

Statistical and Methodological Assessment

The statistical approach using repeated-measures ANOVA is appropriate for the study design. The post-hoc power analysis addresses concerns about sample size adequacy. Effect size reporting enhances the interpretation of practical significance. The randomization procedure appears sound, though more details about allocation concealment would strengthen the methodology section.

7. PLOS authors have the option to publish the peer review history of their article (what does this mean? ). If published, this will include your full peer review and any attached files.

**Do you want your identity to be public for this peer review?** For information about this choice, including consent withdrawal, please see our Privacy Policy .

Reviewer #1: **Yes: ** Wissem Dhahbi

---

## [Author Response · Author response to Decision Letter 2]

2 Sep 2025

We would like to sincerely thank the editor and reviewers for their thorough evaluation of our manuscript and for the constructive feedback provided. Your insightful comments have greatly helped us to refine and strengthen the methodological and statistical aspects of our study. In response to your assessments, we provide the following clarifications and revisions to further enhance the transparency and rigor of our work.

---

## [Editor Report · Decision Letter 2]

4 Sep 2025

Effects of sprint interval training compared to high intensity interval training on repeated sprint capacity and sport-specific performance in college-aged male tennis players

PONE-D-25-21436R2

Dear Dr. Sun,

We’re pleased to inform you that your manuscript has been judged scientifically suitable for publication and will be formally accepted for publication once it meets all outstanding technical requirements.

Kind regards,

Fenghua Sun

Academic Editor

PLOS ONE
---

## [Editor Report · Acceptance letter]

PONE-D-25-21436R2

PLOS ONE

Dear Dr. Sun,

I'm pleased to inform you that your manuscript has been deemed suitable for publication in PLOS ONE. Congratulations! Your manuscript is now being handed over to our production team.

Kind regards,

on behalf of

Dr. Fenghua Sun

Academic Editor

PLOS ONE